# About the Entropy of a Natural Number and a Type of the Entropy of an Ideal

**DOI:** 10.3390/e25040554

**Published:** 2023-03-24

**Authors:** Nicuşor Minculete, Diana Savin

**Affiliations:** Faculty of Mathematics and Computer Science, Transilvania University, Iuliu Maniu Street 50, 500091 Braşov, Romania

**Keywords:** entropy, numbers, ideals, ramification theory in algebraic number fields

## Abstract

In this article, we find some properties of certain types of entropies of a natural number. We are studying a way of measuring the “disorder” of the divisors of a natural number. We compare two of the entropies *H* and $\bar{H}$ defined for a natural number. An useful property of the Shannon entropy is the additivity, $H_{S}\left(\mathrm{pq}\right)=H_{S}\left(p\right)+H_{S}\left(q\right)$, where pq denotes tensor product, so we focus on its study in the case of numbers and ideals. We mention that only one of the two entropy functions discussed in this paper satisfies additivity, whereas the other does not. In addition, regarding the entropy *H* of a natural number, we generalize this notion for ideals, and we find some of its properties.

## 1. Introduction and Preliminaries

In information theory, the entropy is defined as a measure of uncertainty. The most used of the entropies is Shannon entropy ($H_{S}$), which is given for a probability distribution $p=\{p_{1},\ldots ,p_{r}\}$; thus,
$$H_{S}\left(p\right)=-\sum_{i=1}^{r}p_{i}\cdot \log p_{i}.$$

An useful property of the Shannon entropy is the additivity, $H_{S}\left(\mathrm{pq}\right)=H_{S}\left(p\right)+H_{S}\left(q\right)$, where $p=\{p_{1},\ldots ,p_{r}\}$, $q=\{q_{1},\ldots ,q_{r}\}$ and $\mathrm{pq}=\{p_{1}q_{1},\ldots ,p_{1}q_{r},\ldots ,p_{r}q_{1},\ldots ,p_{r}q_{r}\}$.

In [1], Sayyari gave an extension of Jensen’s discrete inequality considering the class of uniformly convex functions getting lower and upper bounds for Jensen’s inequality. He applied this results in information theory and obtained new and strong bounds for Shannon’s entropy of a probability distribution. Recently, in [2], De Gregorio, Sánchez and Toral defined the block entropy (based on Shannon entropy), which can determine the memory for systems modeled as Markov chains of arbitrary finite order.

We have found several ways to define the entropy of a natural number. Jeong et al., in [3], defined the additive entropy of a natural number in terms of the additive partition function. If *d* is the divisor of a natural number *n*, then we will write d|n. If σ(n) is the sum of natural divisors of *n*, then it is easy to see that $\sum_{d|n}^{}\frac{d}{\sigma (n)}=1$. Thus, the ratio $\frac{d}{\sigma (n)}$ can be seen as a probability. As a result we, have a discrete probability distribution associated with a natural number. In [4], we found the following definition for the entropy of a natural number:$$\bar{H}\left(n\right):=-\sum_{d|n}^{}\frac{d}{\sigma (n)}\log\frac{d}{\sigma (n)}=\log\sigma \left(n\right)-\frac{1}{\sigma (n)}\sum_{d|n}^{}d\log d,$$
where log is the natural logarithm. Unfortunately, we did not find this interesting definition of the entropy of a natural number in a book or paper, but on a website. This entropy has the following interesting property:$$\bar{H}\left(mn\right)=\bar{H}\left(m\right)+\bar{H}\left(n\right),$$
when $m,n\in N^{*}$ and gcd(m,n)=1. If *p* is a prime number and $\alpha \in N^{*}$, then we have
$$\bar{H}\left(p^{\alpha }\right)=-\frac{(\alpha +1)\log p}{p^{\alpha +1}-1}+\log\frac{1-p^{-(\alpha +1)}}{p-1}+\frac{p\log p}{p-1}.$$

Taking the limit as α→∞, we obtain
(1)$$\lim_{\alpha \to \infty }\bar{H}\left(p^{\alpha }\right)=\frac{p\log p}{p-1}-\log\left(p-1\right).$$

We remark that, if *p* is a prime number, q>1, such that $\frac{1}{p}+\frac{1}{q}=1$, then
$$H_{S}\left(\frac{1}{p},\frac{1}{q}\right)=\frac{p-1}{p}\left(\frac{p\log p}{p-1}-\log\left(p-1\right)\right)=\left(1-\frac{1}{p}\right)\lim_{\alpha \to \infty }\bar{H}\left(p^{\alpha }\right).$$

In the paper [5], Minculete and Pozna introduced the notion of entropy of a natural number in another way—namely, if n∈N,
n≥2, by applying the fundamental theorem of arithmetic, *n* is written uniquely $n=p_{1}^{\alpha_{1}}p_{2}^{\alpha_{2}}\ldots p_{r}^{\alpha_{r}}$, where $r\in N^{*}$, $p_{1},p_{2},\ldots ,p_{r}$ are distinct prime positive integers and $\alpha_{1},\alpha_{2},\ldots ,\alpha_{r}\in N^{*}.$ Let $\Omega \left(n\right)=\alpha_{1}+\alpha_{2}+\ldots +\alpha_{r}$ and $p\left(\alpha_{i}\right)=\frac{\alpha_{i}}{\Omega \left(n\right)},$ $\left(\forall \right)$
$i=\bar{1,r}$. The entropy of *n* is defined by
$$H\left(n\right)=-\sum_{i=1}^{r}p\left(\alpha_{i}\right)\cdot \log\;p\left(\alpha_{i}\right).$$

Here, by convention, H(1)=0.

Minculete and Pozna (in [5]) gave an equivalent form for the entropy of n, namely:(2)$$H\left(n\right)=\log\;\Omega \left(n\right)-\frac{1}{\Omega \left(n\right)}\cdot \sum_{i=1}^{r}\alpha_{i}\cdot \log\;\alpha_{i}.$$

For example, if n=6=2·3, we have:$$H\left(6\right)=\log\;2-\frac{1}{2}\cdot 2\cdot \log\;1=\log\;2=0.6931\ldots .$$

Another example: if $n=24=2^{3}\cdot 3,$ we have:$$H\left(24\right)=\log\;4-\frac{1}{4}\cdot 3\cdot \log\;3=\frac{1}{4}\cdot \log\left(\frac{4^{4}}{3^{3}}\right)=2.2493\ldots .$$

Minculete and Pozna proved (in [5]) the following:

**Proposition** **1.**

(3)
$$0\leq H\left(n\right)\leq \log\;\omega \left(n\right),\;\left(\forall \right)\;n\in N,\;n\geq 2,$$

*where $\omega \left(n\right)$ is the number of distinct prime factors of n.*


**Remark** **1.**
*(i) If $n=p^{\alpha }$, then $H\left(n\right)=0;$*
*(ii)* 
*If $n=p_{1}\cdot p_{2}\cdot \ldots \cdot p_{r},$ then $H\left(n\right)=log\;\omega \left(n\right);$*
*(ii)* 
*If $n={\left(p_{1}\cdot p_{2}\cdot \ldots \cdot p_{r}\right)}^{k},$ then $H\left(n\right)=log\;\omega \left(n\right).$*



It is easy to see that $H\left(n^{\alpha }\right)=H\left(n\right)$, with α≥1.

The relevance of this entropy is given by the possibility of extension to ideals. The extension of some properties of the natural numbers to ideals was recently given in [6]. Some of the studied results can be transferred to other types of generalized entropies that can be defined later [7]. Entropy is generally used in mathematical physics applications, but it can constitute a new element of analysis in theoretical fields [8]. Recently, in [9], Niepostyn and Daszczuk used entropy as a measure of consistency in software architecture. Therefore, the area of studying different types of entropies in various fields is expanding.

Our motivation of this article was to study some properties of certain types of entropies of a natural number. We compare two of the entropies defined for a natural number. Additionally, regarding the entropy *H* of a natural number, introduced in [5], we generalize this notion for ideals, and we find some of its properties. We mention that the entropy of the ideal is generalized from the second notion of the entropy of integers.

## 2. A Comparison between the Entropies ***H*** and **$\bar{H}$**

In this section, we propose to compare the entropies *H* and $\bar{H}$, looking to similarities and differences between them.

**Proposition** **2.**

(4)
$$\lim_{p\to \infty }\lim_{\alpha \to \infty }\bar{H}\left(p^{\alpha }\right)=0.$$



**Proof.** From relation (Equation 1), we have $\lim_{\alpha \to \infty }\bar{H}\left(p^{\alpha }\right)=\frac{p\log p}{p-1}-\log\left(p-1\right).$ Next, we use the following limit of functions:
$$\lim_{x\to \infty }\left(\frac{x\log x}{x-1}-\log\left(x-1\right)\right)=\lim_{x\to \infty }\frac{x\log x-(x-1)\log(x-1)}{x-1}$$
$$=\lim_{x\to \infty }\left(\log x-\log(x-1)\right)=\lim_{x\to \infty }\log\frac{x}{x-1}=0.$$Therefore, we obtain $\lim_{p\to \infty }\lim_{\alpha \to \infty }\bar{H}\left(p^{\alpha }\right)=\lim_{p\to \infty }\left(\frac{p\log p}{p-1}-\log\left(p-1\right)\right)=0.$ □

**Remark** **2.**
*Related to the entropy $\bar{H}$, we have*

$$\lim_{\alpha \to \infty }\bar{H}\left(np^{\alpha }\right)=\bar{H}\left(n\right)+\frac{p\log p}{p-1}-\log\left(p-1\right),$$

*when gcd(n,p)=1, with p being a prime number and $n,\alpha \in N^{*}$.*

*It is easy to see that $\lim_{p\to \infty }\lim_{\alpha \to \infty }\bar{H}\left(p^{\alpha }\right)=0=H\left(p^{\alpha }\right).$*


**Proposition** **3.**
*If gcd(n,p)=1, with p being a prime number and $n,\alpha \in N^{*}$, then we have*

(5)
$$\lim_{\alpha \to \infty }H\left(np^{\alpha }\right)=0.$$



**Proof.** From the definition of *H*, we have
$$\begin{array}{cc} H(np^{\alpha }) & =\log\left(\Omega \left(n\right)+\alpha \right)-\frac{1}{\Omega (n)+\alpha }\left(\sum_{i=1}^{r}\alpha_{i}\cdot \log\;\alpha_{i}+\alpha \log\alpha \right)\\ \; & =\log\left(\Omega \left(n\right)+\alpha \right)-\frac{\alpha \log\alpha }{\Omega (n)+\alpha }-\frac{1}{\Omega (n)+\alpha }\left(\Omega (n)\log\Omega (n)-\Omega (n)H(n)\right)\\ \; & =\frac{\Omega (n)H(n)}{\Omega (n)+\alpha }+\log\left(\Omega \left(n\right)+\alpha \right)-\frac{\Omega (n)\log\Omega (n)+\alpha \log\alpha }{\Omega (n)+\alpha }. \end{array}$$It follows that
(6)$$H\left(np^{\alpha }\right)=\frac{\Omega (n)H(n)}{\Omega (n)+\alpha }+\log\left(\Omega \left(n\right)+\alpha \right)-\frac{\Omega (n)\log\Omega (n)+\alpha \log\alpha }{\Omega (n)+\alpha }.$$By taking the limit when α→∞, we deduce the relation of the statement. □

We also see that if gcd(m,n)=1, then
H(mn)≠H(m)+H(n).

As a result, we ask ourselves the question of what is the relationship between H(mn) and H(m)+H(n), where $m,n\in N^{*}$, m,n≥2.

If m=22 and n=105, then H(m)=log2, H(n)=log3 and H(mn)=log5, so we have
H(mn)<H(m)+H(n).

If m=20 and n=63, then $H\left(m\right)=H\left(n\right)=\log3-\frac{2}{3}\log2$ and $H\left(mn\right)=\log6-\frac{2}{3}\log2$, which means that
$$H\left(mn\right)-H\left(m\right)-H\left(n\right)=\frac{1}{3}\left(5\log2-3\log3\right)=\frac{1}{3}\log\frac{32}{27}>0,$$
so we have
H(mn)>H(m)+H(n).

Next, we study a general result of this type for the entropy *H*.

**Proposition** **4.**
*We assume that $m=p^{k}q$ and $n=p^{k}t$, where p,q,t are distinct prime numbers and $k\in N^{*}$. Then, the inequality*

H(mn)<H(m)+H(n)

*holds.*


**Proof.** From the definition of *H*, we have $H\left(m\right)=H\left(n\right)=\log\left(k+1\right)-\frac{k}{k+1}\log k$ and $H\left(mn\right)=\log2\left(k+1\right)-\frac{k}{k+1}\log2k$. Therefore, we obtain
$$H\left(m\right)+H\left(n\right)-H\left(mn\right)=\frac{1}{k+1}\left((k+1)\log(k+1)-k\log k-\log2\right).$$We consider the function f:[1,∞)→R defined byf(x)=(x+1)log(x+1)−xlogx−log2. Since $f^{\prime }\left(x\right)=\log\frac{x+1}{x}>0$ for every x≥1, we deduce that the function *f* is increasing, so we have f(x)≥f(1)=log2>0. Consequently, the inequality of the statement is true. □

**Proposition** **5.**
*We assume that $m=p_{1}^{k}p_{2}$ and $n=q_{1}^{k}q_{2}$, where $p_{1},p_{2},q_{1},q_{2}$ are distinct prime numbers and $k\in N^{*}$. Then, we have the following inequality*

H(mn)≥H(m)+H(n).


*Equality holds for k=1.*


**Proof.** For k=1, we deduce that $m=p_{1}p_{2}$ and $n=q_{1}q_{2}$, which implies H(m)=H(n)=log2 and H(mn)=log4, so we have
H(mn)=H(m)+H(n).For k≥2, we find $H\left(m\right)=H\left(n\right)=\log\left(k+1\right)-\frac{k}{k+1}\log k$ and $H\left(mn\right)=\log2\left(k+1\right)-\frac{k}{k+1}\log k$. Now, we obtain
$$H\left(mn\right)-H\left(m\right)-H\left(n\right)=\frac{1}{k+1}\left((k+1)\log2+k\log k-(k+1)\log(k+1)\right)$$
for all k≥2, because the function f:[2,∞)→R defined by f(x)=(x+1)log2+xlogx−(x+1)log(x+1) is strictly positive. It is easy to see that $f^{\prime }\left(x\right)>0$ for every x≥2. Therefore, for x=k, we prove the relation of the statement. □

We study another result for which we have
H(mn)≥H(m)+H(n),
where $m,n\in N^{*}$, m,n≥2.

**Proposition** **6.**
*Let m,n be two natural numbers such that gcd(m,n)=1 and decomposition in prime factors of m,n given by $m=\prod_{i=1}^{r}p_{i}^{a_{i}}$ and $n=\prod_{j=1}^{s}q_{j}^{b_{j}}$ with $a_{i},b_{j}\geq k$ for all i∈{1,…,r} and j∈{1,…,s}, $k\in N^{*}$. Then, the inequality*

$$H\left(mn\right)>H\left(m\right)+H\left(n\right)+\log\left(\frac{k}{\Omega (m)}+\frac{k}{\Omega (n)}\right)$$

*holds.*


**Proof.** Using the definition of *H*, we deduce the equality
(7)$$H\left(mn\right)-H\left(m\right)-H\left(n\right)=\frac{\Omega (n)}{\Omega (m)(\Omega (m)+\Omega (n))}\sum_{i=1}^{r}a_{i}\log a_{i}$$
$$+\frac{\Omega (m)}{\Omega (n)(\Omega (m)+\Omega (n))}\sum_{j=1}^{s}b_{j}\log b_{j}-\log\frac{\Omega (m)\Omega (n)}{\Omega (m)+\Omega (n)}.$$Since $\log a_{i},\log b_{j}\geq \log k$ for all i∈{1,…,r} and j∈{1,…,s}, we obtain that $\sum_{i=1}^{r}a_{i}\log a_{i}\geq \log k\sum_{i=1}^{r}a_{i}=\left(\log k\right)\Omega \left(m\right)$ and $\sum_{j=1}^{s}b_{j}\log b_{j}\geq \log k\sum_{j=1}^{s}b_{j}=\left(\log k\right)\Omega \left(n\right)$. Using equality (Equation 7) and above inequalities, we show that
$$H\left(mn\right)-H\left(m\right)-H\left(n\right)\geq \log k-\log\frac{\Omega (m)\Omega (n)}{\Omega (m)+\Omega (n)}.$$Consequently, the inequality of the statement is true. □

**Theorem** **1.**
*Let m,n be two natural numbers such that gcd(m,n)=1 and H(m),H(n)≥log2. Then, the following inequality*

H(m)+H(n)≥H(mn)

*holds.*


**Proof.** Using relation (Equation 7) and the definition of *H*, we have
(8)$$H\left(mn\right)-H\left(m\right)-H\left(n\right)=\frac{\Omega (n)}{\Omega (m)+\Omega (n)}\left(\log(\Omega (m))-H(m))\right)$$
$$+\frac{\Omega (m)}{\Omega (m)+\Omega (n)}\left(\log(\Omega (n))-H(n)\right)-\log\frac{\Omega (m)\Omega (n)}{\Omega (m)+\Omega (n)}$$
$$=\frac{\Omega (n)\log(\Omega (m))+\Omega (m)\log(\Omega (n))}{\Omega (n)+\Omega (m)}-\frac{\Omega (n)H(m)+\Omega (m)H(n)}{\Omega (n)+\Omega (m)}-\log\frac{\Omega (m)\Omega (n)}{\Omega (m)+\Omega (n)}.$$Since, using the concavity of the function log, we deduce the inequality
$$\frac{\Omega (n)\log(\Omega (m))+\Omega (m)\log(\Omega (n))}{\Omega (n)+\Omega (m)}\leq \log\frac{2\Omega (m)\Omega (n)}{\Omega (m)+\Omega (n)}.$$Therefore, relation (Equation 8) becomes
$$H\left(mn\right)-H\left(m\right)-H\left(n\right)\leq \log2-\frac{\Omega (n)H(m)+\Omega (m)H(n)}{\Omega (n)+\Omega (m)},$$
so we obtain
(9)$$H\left(m\right)+H\left(n\right)-H\left(mn\right)\geq \frac{\Omega (n)H(m)+\Omega (m)H(n)}{\Omega (n)+\Omega (m)}-\log2.$$Therefore, taking into account that H(m),H(n)≥log2 and using inequality (Equation 9), we deduce the statement. □

Next, our goal was to show that the entropy *H* is more suitable to extend it to ideals.

## 3. The Entropy of an Ideal

In this section, we introduce the notion of entropy of an ideal of a ring of algebraic integers, and we find interesting properties of it.

Let *K* be an algebraic number field of degree [K:Q]=n, where n∈N, n≥2, and let $O_{K}$ be its ring of integers. Let Spec$\left(O_{K}\right)$ be the set of the prime ideals of the ring $O_{K}.$ Let *p* be a prime positive integer. Since $O_{K}$ is a Dedekind ring, applying the fundamental theorem of Dedekind rings, the ideal $pO_{K}$ is written uniquely (except for the order of the factors) like this:$$pO_{K}=P_{1}^{e_{1}}\cdot P_{2}^{e_{2}}\cdot \ldots \cdot P_{g}^{e_{g}},$$
where $g\in N^{*},$
$e_{1},e_{2},\ldots ,e_{g}\in N^{*}$ and $P_{1},$
$P_{2},$ …, $P_{g}\in Spec\left(O_{K}\right).$ The number $e_{i}$ ($i=\bar{1,g}$) is called the ramification index of *p* at the ideal $P_{i}$.

Generally, according to the fundamental theorem of Dedekind rings, any ideal *I* of the ring $O_{K}$ decomposes uniquely:(10)$$I=P_{1}^{e_{1}}\cdot P_{2}^{e_{2}}\cdot \ldots \cdot P_{g}^{e_{g}},\;where\;r\in N^{*},e_{1},e_{2},\ldots ,e_{g}\in N^{*}\;and\;P_{1},P_{2},\ldots ,P_{g}\in Spec\left(O_{K}\right).$$
We shall mostly work in this article with ideals of the form $pO_{K}$, since for such ideals there are known ramification results in the ring $O_{K}$, for many algebraic number fields *K* (when *K* is any quadratic field, or *K* is any cubic field, or *K* is any cyclotomic field, or *K* is any Kummer field, etc.)

The following result is known (see [10,11,12]):

**Proposition** **7.**
*In the above notation, we have:*
*(i)* 

$$\sum_{i=1}^{g}e_{i}f_{i}=\left[K:Q\right]=n,$$

*where $f_{i}$ is the residual degree of p, meaning $f_{i}=\left[O_{K}/P_{i}:Z/pZ\right],$
$i=\bar{1,g}.$*
*(ii)* 
*If, moreover, Q⊂K is a Galois extension, then $e_{1}=e_{2}=\ldots =e_{g}$ (denoted by e), $f_{1}=f_{2}=\ldots =f_{g}$ (denoted by f). Therefore, efg=n.*



Let J be the set of ideals of the ring $O_{K}.$ Let *I*∈J, *I* be written uniquely as in equality (Equation 10).

It is easy to see that $\sum_{i=1}^{g}\frac{e_{i}}{\Omega (I)}=1$. Thus, the ratio $\frac{e_{i}}{\Omega (I)}$ can be seen as a probability; as a result, we have a discrete probability distribution associated with a ideal.

We generalize the notion of entropy of an ideal like this:

**Definition** **1.**
*Let $I\neq \left(0\right)$ be an ideal of the ring $O_{K}$, decomposed as above. We define the entropy of the ideal I as follows:*

(11)
$$H\left(I\right)=-\sum_{i=1}^{g}\frac{e_{i}}{\Omega (I)}\log\;\frac{e_{i}}{\Omega (I)},$$

*where $\Omega \left(I\right)=e_{1}+e_{2}+\ldots +e_{g}.$*


Immediately, we obtain the following equivalent form, for the entropy of the ideal *I*:(12)$$H\left(I\right)=\log\;\Omega \left(I\right)-\frac{1}{\Omega \left(I\right)}\cdot \sum_{i=1}^{g}e_{i}\cdot \log\;e_{i}.$$

We now give some examples of calculating the entropy of an ideal.

**Example** **1.**
*Let ξ be a primitive root of order 5 of the unity and let $K=Q\left(\xi \right)$ be the 5th cyclotomic field. The ring of algebraic integers of the field K is $O_{K}=Z\left[\xi \right].$ We consider the ideal $\left(1-\xi \right)\cdot Z\left[\xi \right].$ It is known that $\left(1-\xi \right)\cdot Z\left[\xi \right]\in Spec\left(O_{K}\right)$ (see [10,13]). Let the ideal $5\cdot Z\left[\xi \right]={\left(1-\xi \right)}^{4}\cdot Z\left[\xi \right].$ The entropy of the ideal $5\cdot Z\left[\xi \right]$ is*

$$H\left(5\cdot Z\left[\xi \right]\right)=\log\;4-\frac{1}{4}\cdot 4\cdot \log\;4=0.$$



**Example** **2.**
*Let the pure cubic field $K=Q\left(\sqrt[{3}]{2}\right).$ Since $2^{2}≢1$ (mod 9), the results show that the ring of algebraic integers of the field K is $O_{K}=Z\left[\sqrt[{3}]{2}\right]$ (see [14]).*

*Since 29≡2 (mod 3), $29Z\left[\sqrt[{3}]{2}\right]=P_{1}\cdot P_{2},$ where $P_{1},P_{2}\in Spec\left(Z\left[\sqrt[{3}]{2}\right]\right).$ Thus, the ideal $29Z\left[\sqrt[{3}]{2}\right]$ splits in the ring $Z\left[\sqrt[{3}]{2}\right].$ The entropy of the ideal $29Z\left[\sqrt[{3}]{2}\right]$ is*

$$H\left(29Z\left[\sqrt[{3}]{2}\right]\right)=\log\;2-\frac{1}{2}\cdot 2\cdot \log\;1=\log\;2.$$



**Example** **3.**
*In the same field (as in the previous example) $K=Q\left(\sqrt[{3}]{2}\right)$ with the ring of integer $O_{K}=Z\left[\sqrt[{3}]{2}\right],$ we consider the ideal $31Z\left[\sqrt[{3}]{2}\right].$*

*Since 31≡1 (mod 3), $31Z\left[\sqrt[{3}]{2}\right]=P_{1}\cdot P_{2}\cdot P_{3},$ where $P_{1},P_{2},P_{3}\in Spec\left(Z\left[\sqrt[{3}]{2}\right]\right).$ Thus, the ideal $31Z\left[\sqrt[{3}]{2}\right]$ splits completely in the ring $Z\left[\sqrt[{3}]{2}\right]$ (see [14]). The entropy of the ideal $31Z\left[\sqrt[{3}]{2}\right]$ is*

$$H\left(31Z\left[\sqrt[{3}]{2}\right]\right)=\log\;3-\frac{1}{3}\cdot 3\cdot \log\;1=\log\;3.$$



**Remark** **3.**
*Let K be an algebraic number field, and let $O_{K}$ be its ring of integers. Let p be a prime positive integer. If p is inert or totally ramified in the ring $O_{K},$ then $H\left(pO_{K}\right)=0.$*


**Proof.** To calculate the entropy of ideal $pO_{K}$, we consider two cases.**Case 1**: if *p* is inert in the ring $O_{K},$ the results show that $pO_{K}$ is a prime ideal. Then $\Omega \left(pO_{K}\right)=1$ and $H\left(pO_{K}\right)=0.$**Case 2**: if *p* is totally ramified in the ring $O_{K},$ the results show that $pO_{K}=P^{n},$ where $P\in Spec\left(O_{K}\right)$ and n=[K:Q]. This results immediately in $\Omega \left(pO_{K}\right)=n$ and $H\left(pO_{K}\right)=log\;n-log\;n=0.$ □

**Proposition** **8.**
*Let n be a positive integer, n≥2, and let p be a positive prime integer. Let K be an algebraic number field of degree [K:Q]=n and let $O_{K}$ be its ring of integers. Then:*

(13)
$$0\leq H\left(pO_{K}\right)\leq \log\;\omega \left(pO_{K}\right)\leq \log\;n,\;$$

*where $\omega \left(pO_{K}\right)$ is the number of distinct prime factors of the ideal $pO_{K}.$*


**Proof.** The proof of the inequality $0\leq H\left(pO_{K}\right)\leq \log\;\omega \left(pO_{K}\right)$ is similar to the proof of Proposition 1 (that is, Theorem 2. from the article [5]).Since $O_{K}$ is a Dedekind ring, the ideal $pO_{K}$ is written in a unique way:
$$pO_{K}=P_{1}^{e_{1}}\cdot P_{2}^{e_{2}}\cdot \ldots \cdot P_{g}^{e_{g}},$$
where $g\in N^{*},$
$e_{1},e_{2},\ldots ,e_{g}\in N^{*}$ and $P_{1},$
$P_{2},$…, $P_{g}\in Spec\left(O_{K}\right).$ By applying Proposition 7 (i), we obtain that $\omega \left(pO_{K}\right)=g\leq n.$ The equality $\omega \left(pO_{K}\right)=n$ is achieved when the ideal *p* splits totally in the ring $O_{K}.$ It follows that
$$0\leq H\left(pO_{K}\right)\leq log\;\omega \left(pO_{K}\right)\leq log\;n.$$□

**Proposition** **9.**
*Let K be an algebraic number field, and let $O_{K}$ be its the ring of integers. Let p be a prime positive integer. If the extension of fields Q⊂K is a Galois extension, then*

$$H\left(pO_{K}\right)=\log\;\omega \left(pO_{K}\right).$$



**Proof.** By taking into account the fact that $O_{K}$ is a Dedekind ring and applying Proposition 7 (ii), it follows that the ideal $pO_{K}$ is uniquely written as follows:
$$pO_{K}=P_{1}^{e_{1}}\cdot P_{2}^{e_{1}}\cdot \ldots \cdot P_{g}^{e_{1}},$$
where $g\in N^{*},$
$e_{1}\in N^{*}$ and $P_{1},$
$P_{2},$…, $P_{g}\in Spec\left(O_{K}\right).$ According to Formula (2), the entropy of the ideal $pO_{K}$ is
$$H\left(pO_{K}\right)=\log\left(ge_{1}\right)-\frac{1}{ge_{1}}\cdot ge_{1}\cdot \log\;e_{1}=\log\;g=\log\;\omega \left(pO_{K}\right).$$□

## 4. Conclusions

Study of the entropy in information theory is a very important tool for for measuring uncertainty. The most used of entropies is the Shannon entropy. There are many studies regarding the characterization and application of entropy Shannon (see, e.g., [1,2], etc.). We are studying a way of measuring the “disorder” of the divisors of a natural number. Since we have $\sum_{d|n}^{}\frac{d}{\sigma (n)}=1$, the ratio $\frac{d}{\sigma (n)}$ can be seen as a probability. As a result, we have a discrete probability distribution associated with a natural number. Similarly, there are some studies related to the entropy of a natural number—namely, Jeong et al., in [3], defined the additive entropy of a natural number in terms of the additive partition function, and in [4], we found the following definition for the entropy of a natural number:$$\bar{H}\left(n\right):=-\sum_{d|n}^{}\frac{d}{\sigma (n)}\log\frac{d}{\sigma (n)}=\log\sigma \left(n\right)-\frac{1}{\sigma (n)}\sum_{d|n}^{}d\log d,$$
where σ(n) is the sum of natural divisors of *n*. Additionally, regarding the entropy *H* of a natural number, introduced in [5], another type of entropy is a natural number. Mainly, the discussion is about the properties of entropy *H*. In Propositions 6 and Theorem 1, we were talking about the magnitude of H(mn) and H(m)+H(n).

In equality $\sum_{i=1}^{g}\frac{e_{i}}{\Omega (I)}=1$, the ratio $\frac{e_{i}}{\Omega (I)}$ can be seen as a probability. As a result, we have a discrete probability distribution associated with a ideal. Thus, we generalize this notion for ideals and find some of its properties. The relation between the proposed entropy of a natural number or an ideal is of a purely theoretical nature.

In the future, we will look for other connections of entropy within ideals, studying a possible generalization of existing entropy types for natural numbers or for ideals. We will study some inequalities involving the entropy *H* of an exponential divisor of a positive integer and the entropy *H* of an exponential divisor of an ideal. Additionally, we shall try to study the entropy in the cases of more general ideals of the ring of algebraic integers $O_{K}$ of an algebraic number field K, than the ideals of the form $pO_{K},$ with *p* being a prime integer.

## Data Availability

Not applicable.

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
