# Peer review of "About the Entropy of a Natural Number and a Type of the Entropy of an Ideal"

_entropy, 2023, doi:10.3390/e25040554_

Round 1
Reviewer 1 Report
This paper discusses the entropy of a natural number with two different perspectives and, in addition, the entropy of an ideal with some properties. The entire paper is based on the mathematical derivations and it is no where explained the implications or applications of such properties. Hence, I suggest the authors to provide the clear motivations and applications of derived properties. Moreover, please rewrite the abstract; it is quite confusing to the readers.
Reviewer 2 Report
(1) Some expressions need to be improved.
(2) In introduction, some other relevant work may be referenced and the citation needs to be in certain order. Some symbols such as “d” in the second equation and the equation should be explained. Relevant real problems and connections to other work of the proposed entropy may be introduced. For example, the “p” in the conventional entropy represents probability which is less than 1, and what is the relation between the proposed entropy of natural numbers with the conventional? The “log” below equation (1.1) should be given before. The number (1.1) of equations was used repeatedly.
(3) Please check equations with expressions, for example, Remark 2. Section 2 was given to intend to compare the two entropies. However, from Remark 3, the discussions were on the conventional entropy. Could some discussions on the proposed entropy of natural numbers be added?
(4) In Section 3, the discussions were based on the conventional entropy generalized. Could the discussions on the proposed entropy of natural numbers or some connections between them be added?
(5) Expressions in Conclusions need to be revised.
Round 2
Reviewer 1 Report
Accept
Author Response
Thank you for your review and comments